# Origin and Fate of Acrolein in Foods

**DOI:** 10.3390/foods11131976

**Published:** 2022-07-03

**Authors:** Kaiyu Jiang, Caihuan Huang, Fu Liu, Jie Zheng, Juanying Ou, Danyue Zhao, Shiyi Ou

**Affiliations:** 1Department of Food Science and Engineering, Jinan University, Guangzhou 510632, China; kaiyujiang@stu2020.jnu.edu.cn (K.J.); thamy@jnu.edu.cn (C.H.); liu_fu@jnu.edu.cn (F.L.); zhengjie@jnu.edu.cn (J.Z.); 2Institute of Food Safety & Nutrition, Jinan University, Guangzhou 510632, China; toujy@jnu.edu.cn; 3Research Institute for Future Food, The Hong Kong Polytechnic University, Hong Kong 999077, China; daisydy.zhao@polyu.edu.hk; 4Guangdong-Hong Kong Joint Innovation Platform for the Safety of Bakery Products, Guangzhou 510632, China

**Keywords:** acrolein, conversion, foods, formation, interactions, metabolism, molecular mechanisms

## Abstract

Acrolein is a highly toxic agent that may promote the occurrence and development of various diseases. Acrolein is pervasive in all kinds of foods, and dietary intake is one of the main routes of human exposure to acrolein. Considering that acrolein is substantially eliminated after its formation during food processing and re-exposed in the human body after ingestion and metabolism, the origin and fate of acrolein must be traced in food. Focusing on molecular mechanisms, this review introduces the formation of acrolein in food and summarises both in vitro and in vivo fates of acrolein based on its interactions with small molecules and biomacromolecules. Future investigation of acrolein from different perspectives is also discussed.

## 1. Introduction

Acrolein is derived from the thermal degradation of glycerine, a by-product of soap manufacturing; it was first named by Brandes because of its acrid smell and oil-like viscosity [1]. In 1839, Berzelius characterised acrolein as an aldehyde [2].

Acrolein is a structurally simple α,β-unsaturated aldehyde [3] present in the environment, water and food due to all kinds of human activities, such as high-temperature cooking, fossil fuel combustion, vehicle exhaust emission and insecticide abuse [4]. Human exposure to acrolein primarily occurs through air inhalation, dietary intake and endogenous release. Exogenous or endogenous acrolein can exert deleterious health effects due to its high toxicity. Given its highly electrophilic structure, acrolein can easily bind to some nucleophilic biomacromolecules, such as protein and nucleic acids. The binding of acrolein to biomacromolecules results in oxidative stress [5], endoplasmic reticulum stress [6], mitochondrial dysfunction [7] or even inflammation [8] and abnormal immune responses [9]. Over the past few decades, numerous studies have investigated the relationship between acrolein and diseases, especially several chronic diseases, including cardiovascular disease [10], alcoholic liver disease [11], Alzheimer’s disease [12], diabetes [13] and chronic obstructive pulmonary disease (COPD) [14]. Furthermore, recent studies have found that acrolein-induced DNA damage contributes to tumourigenesis [15]. As a result of the ubiquitous sources and potential health risks of acrolein, the U.S. Environmental Protection Agency has listed acrolein as a high-priority toxic chemical [16], and the International Programme on Chemical Safety of the World Health Organisation (WHO) has set a tolerable daily acrolein intake level of 7.5 μg/kg bw/day [17].

Dietary intake is one of the main routes of human exposure to acrolein. Acrolein in foods differs substantially in content depending on the kinds of foods and their processing conditions. Interestingly, the content of oil-soluble acrolein in frying oils is much higher than that in fried foods, up to approximately 3700-fold in frying rapeseed oil than in fried doughnuts [18]. A major part of acrolein is eliminated after its formation in processed foods possibly via its interactions with other food components. However, re-exposure of the eliminated acrolein occurs after food intake, and up to a 38-fold increase in internal acrolein has been found in volunteers after an intake of potato crisps [19]. Apart from a discussion of the formation mechanisms of acrolein in foods, this review introduces in vitro and in vivo transformation pathways of acrolein based on its interactions with small molecules and biomacromolecules. We aimed to encourage scientists to examine the largely eliminated part of acrolein produced in foods.

## 2. Origin of Acrolein in Foods

The pervasive occurrence of acrolein in foods is mainly related to frying, baking and fermentation. Acrolein can also be endogenously released by enzymatic reactions or drug metabolism in the body and, more intriguingly, by the fermentation of gut microbiota which was reported in recent studies [20].

### 2.1. Acrolein Content in Foods

As shown in Table 1 [21,22,23,24,25,26,27,28,29,30,31,32], acrolein has been found in a variety of foods, including fruits, vegetables, baked or fried foods and alcoholic beverages. Acrolein levels change erratically during food processing and ingestion. For example, heated rapeseed oil has an acrolein content of approximately 150 mg/kg [26], but potato chips fried in rapeseed oil have an acrolein content of only 23 μg/kg [18]. The amount of acrolein-related metabolites excreted in urine after potato chip consumption is considerably higher than the calculated acrolein content of the consumed chips [19]. Moreover, acrolein is a common carbonyl compound that is present at widely different concentrations in all types of alcoholic beverages. If liquors are considered, daily acrolein intake can exceed 1 mg [10], which is far above the tolerable daily intake value of 7.5 μg/kg bw/day set by the WHO. As a result of different dietary habits and customs, the real acrolein exposure level is difficult to determine and may be underestimated when calculated on the basis of acrolein content in food alone. Hence, the main mechanisms of acrolein generation in food are discussed below to further understand dietary acrolein.

### 2.2. Acrolein Formation from Edible Oils

Numerous studies have reported that acrolein can be formed during the heating of oils and fats. The continuously generated acrolein is retained in heated oil, migrated to processed foods or dispersed into the air. As shown in Figure 1A, during oil heating, water attacks the ester bond of triglycerides and triggers thermal hydrolysis to release glycerol and fatty acids [33]. Acrolein in lipids can also form from the oxidation of polyunsaturated fatty acids in addition to the dehydration of glycerol. In 1987, Umano proposed the formation pathways of acrolein from triglyceride in cooking oils and reported a free radical-initiating pathway [34]. They supposed that acrolein can be produced by the successive homolytic fissions of ester linkages. Different levels of acrolein can be detected in oils with varying fatty acid compositions, and the peroxidation of polyunsaturated fatty acids has been documented to be the main source of acrolein. In 1991, Esterbauer and his colleagues proposed the formation of acrolein from arachidonic acid [35]. Notably, acrolein is generated from the middle of the fatty acid chain, not from the alkyl terminus nor carboxy terminus. In the presence of oxygen, the twice successive production of hydroperoxide and the β-cleavage of alkoxy radicals are needed to generate acrolein. In 2013, Endo et al. suggested that linolenic acid may be the main source of acrolein from heated vegetable oils and described the mechanism of its generation [36]. Specific hydroperoxide and epidioxide are the key intermediates in acrolein formation. By following the idea of Esterbauer’s review, Ewert elucidated a similar mechanism of acrolein formation from linoleic acid [18]. In a recent study, Kato et al. [37] also confirmed that the singlet oxygen oxidation products of linoleic acid and linolenic acid, but not oleic acid, are significant sources of acrolein formation. Moreover, acrolein was confirmed to be the major volatile in the early stages of fish oil oxidation and may be responsible for flavour deterioration [38]. Eicosapentaenoic acid (EPA) and docosahexaenoic acid (DHA), the common ω-3 fatty acids in fish oil, can also be the precursors of acrolein and Shibata inferred their related decomposition mechanism [39]. EPA and DHA are oxidised into hydroperoxide and then rapidly decomposed to form an alkoxyl radical and a hydroxyl radical. Further scission yields a propanal and an alkyl radical, and acrolein is produced by the successive reaction of this alkyl radical with hydroxyl radicals. The cleavage mechanism of polyunsaturated fatty acids to form acrolein is illustrated in Figure 1B.

### 2.3. Acrolein Formation from Carbohydrates

Many carbonyl compounds are formed in foods during thermal processing and contribute to their special flavour and aroma. Carbohydrates, which are one of the main ingredients in thermally processed foods, may be pyrolysed into acrolein (Figure 2). In 1966, Byrne et al. suggested that acrolein and hydroxypyruvaldehyde can be generated from the pyrolysis of hexoses [40]. Paine Ⅲ et al. later supported this idea via isotopic labelling to clarify the pyrolysis mechanism of glucose [41]. They confirmed that acrolein originates from C4 to C6 of the decomposed glucose, and the dominant labelling pattern of acrolein can be rationalised via a two-step pathway. Firstly, the dehydration between C5 and C3 leads to the cleavage of C3 and C4, forms propene-1,3-diol and yields acrolein after further dehydration. Another possible pathway may be dehydration between C1 and C2 and the subsequent retro Diels–Alder reaction on the same diol. In addition, they offered a minor pathway for acrolein formation involving the hydride shift from C6 and the loss of hydroxide from C4. Yaylayan and Stevens reported that hydroxyacetone is the precursor of acrolein and can be formed through the 3,4-retro aldol cleavage of glucose [42,43]. 

In addition to the thermal degradation of sugars, acrolein is produced during specific stages of the Maillard reaction through the mechanisms presented in Figure 3. In its primary stage, Schiff bases are formed by sugars and amino acids (AAs) and easily transferred into Amadori products. Similarly, hydroxyacetone is formed through the dehydration and 3,4-retro aldol cleavage of the Amadori product and may finally generate acrolein. If the amines are lost, acrolein is formed the same way after the formation of 1-deoxyglucosone. Sterecker degradation in the intermediate stage of the Maillard reaction can be another source of acrolein. For example, methionine can react with dicarbonyl compounds and convert into methional. Through a retro Michael addition reaction, acrolein can be released with methyl mercaptan [44]. The occurrence and disappearance of acrolein, a highly reactive aldehyde, co-exists during carbohydrate pyrolysis and the Maillard reaction. Further insights into the formation of acrolein in the thermal processing of food can be obtained through the further clarification of these complicated mechanisms.

### 2.4. Acrolein Formation from Alcoholic Beverages

The consumption of alcoholic beverages appears to be an important way to increase oral exposure to acrolein. Acrolein was first found in wine because of its contribution to bitterness and not for its toxic effects. The bitterness of wine can be caused by phenolic compounds and then relieved with the gradual oxidative polymerisation and precipitation of these phenolics. Pasteur first observed that bitterness in wine develops with the growth of bacteria and the loss of glycerol [45]. In the early 20th century, Voisenet correlated the bitterness of wine with the presence of acrolein through a series of experiments. He found that acrolein was formed from glycerol under fermentation by *Bacillus amaracrylus* [46]. As shown in Figure 4, glycerol can be dehydrated into 3-hydroxypropionaldehyde (3-HPA) by a coenzyme B12-dependent glycerol/diol dehydratase [47] and then converted into acrolein via another dehydration reaction. According to Mills et al., given that the formation of acrolein in distillery mashes decreases with the increasing addition of glucose, conversion into acrolein is restricted when glucose is sufficient as the carbon source for fermentation [48]. Despite the bitter taste of acrolein itself [49], Rentschler et al. confirmed that the condensation of anthocyanins and acrolein contributes to the development of bitterness in wine [50]. Moreover, 3-HPA is not the end product of fermentation. It can be further reduced into 1,3-propanediol by oxidoreductase. The presence of the dimer and hydrated forms of 3-HPA was successively confirmed by Hall and Stern [51] in 1950 and by Nielsen et al. [52] in 1981. These two missing pieces were finally found to complete the HPA system puzzle, which has been concluded by many researchers. Sung et al. reported that more 3-HPA oligomers may occur mainly via acetalisation or aldol reaction depending on different pH conditions [53]. In accordance with this mechanism, 3-HPA and its derivatives in various forms can be the stock of acrolein in wine and transform into acrolein under specific conditions. For example, heating and acidic pH are highly suitable for acrolein formation from 3-HPA and may account for the higher levels of acrolein in cognac than in wine. Bacteria, such as *Lactobacillus*, are the producers of acrolein in wine [54], and the final concentration of acrolein is regulated by the HPA system. Once the brewing process or the storage condition is changed, high amounts of acrolein may be released into wine due to a shift in the reaction equilibrium. Although acrolein was first noticed in wine because it promotes bitterness, the health risk posed by acrolein at a content level exceeding its safety limit should be considered as a highly critical issue that is worthy of concern.

## 3. Fate of Acrolein in Foods

Acrolein has high reactivity because of its α,β-unsaturated carbonyl structure. Acrolein generated through different pathways can act as an active intermediate and react with other substances to transform into highly stable forms. For example, acrolein contributes to the formation of acrylamide during food thermal processing, whereas acetal and dimers are the main forms of acrolein present in alcoholic beverages. The diverse transformation pathways of acrolein account for its low detected levels in foods and may cause the miscalculation of real human exposure to acrolein. Given that acrolein is a highly toxic hazard, many researchers have attempted to eliminate it and mitigate its toxic effects by using food components, natural products and some clinical drugs. These control strategies appear to be the artificial fates designed for acrolein. Furthermore, when acrolein is absorbed by the human body or produced endogenously, partial acrolein can be metabolised and then excreted through urine or exhalations. However, excess acrolein may conjugate to biomacromolecules, induce protein dysfunction and DNA damage and ultimately cause detrimental health effects. In this section, the fates of acrolein are discussed by introducing the interactions of acrolein with small molecules, its conjugation to biomacromolecules and its metabolic pathways in the human body.

### 3.1. Interactions of Acrolein with Small Molecules

#### 3.1.1. Conversion of Acrolein during Food Processing

Acrolein has been detected in various foods and has different conversion pathways depending on processing and storage conditions. For example, although acrolein is considered one of the sources of bitterness in wine and liquor, it can easily react with three molecules of ethanol in acidic medium to form 1,1,3- triethoxypropane [55], which imparts a highly fruity aroma to cider [56,57]. In addition, the dimer form of acrolein was suggested to be a marker for the detection of acrolein in wine [58]. Acrolein generated from heated oil is thought to contribute to the formation of acrylamide in thermally processed foods, such as fried foods. Acrolein may be oxidised and react with ammonia to form acrylamide through the acrylic acid pathway [59]. By using the simulated thermal processing model, Zamora et al. [60] suggested that acrolein can react with creatinine and finally yield 2-amino-3-methylimidazo(4,5-f)quinolone (IQ) and 2-amino-3,8-dimethylimidazo [4,5-f]quinoxaline (MeIQx) [61], which were listed as Group 2A (probable carcinogenic) and Group 2B agents (possibly carcinogenic), respectively, by the International Agency for Research on Cancer. Their results demonstrated that acrolein is responsible for the formation of heterocyclic aromatic amines with the aminoimidazoazarene structure in foods. However, by excluding the contaminants converted from acrolein, Zamora et al. [62] suggested a formation route for 3-picoline from acrolein in foods by using isotopic labelling, which illustrated the role of acrolein in flavour generation. The conversion pathways of acrolein in different foods are depicted in Figure 5.

#### 3.1.2. Interactions of Acrolein with Amino Acids

Acrolein can be scavenged by nucleophilic components in foods (Figure 6). Our research group recently found that AAs can effectively scavenge acrolein; amongst them, cysteine eliminates acrolein with higher efficiency than other AAs due to its thiol group [63]. Cysteine can react with acrolein and form the adduct Acr-di-Cys by Michael addition reaction and cyclodehydration. The adduct Acr-di-Cys shows much lower cytotoxicity than acrolein towards Caco-2 cells and HBE cells, but the cytotoxicity of its decomposed product Acr-mono-Cys is even higher than that of acrolein [64]. We also identified the adducts formed between acrolein and γ-aminobutyric acid (GABA) [65], alanine (Ala) or serine (Ser) [66]. These three AAs can scavenge more than 80% of acrolein under both physiological and thermal processing conditions via the formation of formyl-dehydropiperidino structures. The formation of these adducts can mitigate acrolein-induced cytotoxicity through a reduction in cellular reactive oxygen species and apoptosis. Recently, we reported that AAs can simultaneously react with two reactive carbonyl species (RCS) to form a new type of adduct [67], which suggests that the scavenging mechanism of AA to acrolein may be complicated under co-existence with other RCS.

#### 3.1.3. Interactions of Acrolein with Antioxidant and Natural Products

Furthermore, some common antioxidants applied in foods have been confirmed to act as acrolein scavengers. In 1983, Fodor et al. proposed that L-ascorbic acid is a good Michael donor for reaction with acrolein and that it undergoes enediol-mediated intramolecular cyclisation to form ascorbylated acrolein [68]. Wang et al. found that propyl gallate, a well-known lipid oxidant, can effectively eliminate acrolein at high temperatures and examined its trapping effect on acrolein in real food by using a cake model [69]. Notably, the use of the components or nutrients in food itself as acrolein scavengers has unique advantages in that it may not lead to unpleasant flavours or new health risks from foreign ingredients.

Natural products have been found to have a key role in the control of food-derived hazards, and some have shown great potential in scavenging acrolein. In 2009, Wang et al. compared the acrolein-trapping activity of 21 kinds of polyphenols with different structural characteristics. Their results revealed that the structure of polyphenols affects reactivity to acrolein. They also identified the structure of phloretin–acrolein adducts and concluded that polyphenol–acrolein adducts are formed via the Michael addition reaction and share a common 2-chromanol skeleton [70]. In a paper published in 2012, Wang et al. suggested that phloretin can attenuate acrolein-induced cytotoxicity by sacrificially reacting with acrolein [71]. Polyphenols with a resorcinol structure are considered highly promising agents with the further development of research. Wang et al. found that resveratrol and hesperetin can remove acrolein rapidly under a simulated physiological condition [72]. Koichi et al. reported that the addition of matcha powder can suppress acrolein accumulation during cake baking and detected mono catechin–acrolein adducts in cakes with matcha powder added to them [73]. In recent years, Lv et al. successively published their research on the control of acrolein by polyphenols, such as myricetin [74], epigallocatechin-3-gallate (EGCG), genistein [75] and cyanidin-3-O-glucoside (C3G) [76]. Notably, they firstly confirmed the acrolein-trapping effect of EGCG and genistein in vivo and then identified the formation of adducts in mouse urine. Furthermore, in their recent study, they found that curcumin reacted with acrolein at the site of its active methylene; in grilled chicken wings, the combined addition of quercetin and curcumin showed a synergistic effect to inhibit acrolein [77]. Tao et al. pointed out the unique reaction mechanism between ferulic acid and acrolein in addition to polyphenols. In this mechanism, acrolein induces the direct decarboxylation of ferulic acid and then yields an adduct. Through a structure–reactivity relationship study, Tao et al. suggested that the presence of 4-OH is necessary for decarboxylation [78]. To summarise, although natural products have shown great potential for acting as acrolein scavengers, the health risk of interventions based on natural products should be considered because adduct formation of polyphenol and carbonyl compounds can increase toxicity [79]. However, natural products may potentially mitigate acrolein-induced toxicity not only through their direct scavenging effect but also by regulating related cell signalling pathways. Such an effect may exhibit a dual protection mechanism.

#### 3.1.4. Interactions of Acrolein with Clinical Drugs

As mentioned above, high levels of acrolein can be detected in the pathological tissues of patients. Given that high acrolein levels can cause oxidative stress and carbonyl stress that exacerbate many diseases, the development of drugs targeting acrolein may offer a new strategy for control or treating these diseases. Many common clinical drugs have been used as potent acrolein-sequestering agents, which includes thiol-containing and hydrazine-containing substances [80]. Drugs with thiol groups, such as amifostine, mesna, N-acetylcysteine (NAC) and cysteamine [81,82], show considerable acrolein sequestering activity. Amifostine can be metabolised by alkaline phosphatase to release active WR-1065. Similar to other thiol-containing drugs, WR-1065 can take effect rapidly to form mercaptopropanal via thiol–ene addition. Cysteamine can concurrently trap the carbonyl group of acrolein and form adducts with thiazolidine structures. Hydrazine-containing drugs can react with acrolein to form hydrazone derivatives [83]. For example, compared with hydralazine, dihydralazine can trap more acrolein because it possesses one more hydrazine group [84]. Notably, phenelzine can not only trap acrolein directly, but it can also act as an inhibitor of MAO to prevent the endogenous formation of acrolein [85]. In particular, edaravone can target the ene and carbonyl groups of acrolein to yield di-adducts [86]. Moreover, caffeine and theophylline [87] can trap acrolein, and their metabolites can continually scavenge acrolein in vivo; these adducts could be detected in the urinary samples of volunteers [88]. In conclusion, although many clinical drugs have shown effective acrolein-scavenging activity in a large amount of studies and have even been clinically applied to avoid acrolein-induced toxic side effects, some studies have indicated that the accumulation of drug–acrolein adducts may trigger new health risks [89]. Therefore, the dosages of these drugs should be carefully considered to obtain a balanced benefit:risk ratio. The pharmaceutical fates of adducts formed through drug–acrolein interactions (or drug–acrolein–protein interactions) remain unclear and should be further investigated. In this way, we can select the appropriate dose of medicine to maximise its curative effects on acrolein-related diseases.

Taken together, the interactions of acrolein, the simplest α,β-unsaturated aldehyde with high reactivity, with small molecules are complicated and remain in constant development. Although acrolein is acclaimed for its facile conversion into many industrial manufacturing ingredients, its high toxicity remains a concern. Given the polytropic transformation pathways and multiple fates of acrolein in different environments, the detectable part of acrolein may only be a fraction of the real level of acrolein. Therefore, the interaction mechanisms of acrolein and other small molecules under diverse conditions must be elucidated. Such information may help establish reliable risk assessment methods for acrolein in accordance with its existing environment. The discovery and application of some acrolein-scavenging agents appears to be necessary in the design of artificial fates for acrolein to avoid acrolein-induced toxic effects. Nevertheless, their potent acrolein-trapping activity is inadequate for proving their feasibility because their safety is unclear. Therefore, the health risks of adducts formed by these trapping reactions must be emphasised after the identification of these agents and their metabolic pathways need further research to enable the precise, effective and safe control of acrolein in different cases.

### 3.2. Conjugations of Acrolein to Biomacromolecules

The electron-deficient structure of acrolein facilitates its reaction with cellular nucleophiles, such as proteins and DNA. The mechanism of acrolein toxicity is known to be frequently related to protein modification and DNA adduction. Acrolein-induced protein modifications can significantly alter protein function and affect enzyme activity or cell signalling, whereas acrolein–DNA adduction may cause mutations and epigenetic modifications. Both of these processes can lead to disease states in various biological tissues and then trigger the development of related diseases. Hence, summarising the conjugations of acrolein to these biomacromolecules will help identify acrolein-preferring acting sites and clarify the causal relationship between acrolein and pathogenesis. Such information may provide an indication as to how to prevent acrolein-related diseases by protecting the acrolein target. This section focuses on the conjugations of acrolein to protein and DNA, as well as the results of relevant studies in recent years.

#### 3.2.1. Conjugations of Acrolein to Peptides and Proteins

Given that proteins comprise specific AAs, protein modifications actually result from reactions between acrolein and AA residues. Uchida et al. conducted numerous studies on conjugation between acrolein and protein AA residues. In 1998, they first reported that acrolein can react with the lysine residue of oxidised LDL via Michael addition on its ε-amino group and aldol reaction to form FDP–lysine [90], which has been used as a biomarker of acrolein-induced protein modification [91]. In a further study, they identified another acrolein–lysine residue adduct with a pyridinium structure that formed from a Schiff base product [92]. They also observed the reaction product of acrolein and histidine residues formed via the Michael addition on the imidazole group [93]. According to Seiner et al., in addition to reacting with nitrogen-containing groups, acrolein can inactivate protein tyrosine phosphatase 1B via conjugation on the active site cysteine-215 [94]. Cai et al. proposed the mechanism underlying the transfer of Schiff base products between peptides derived from the formation of the Michael-type adduct of cysteine and acrolein [95]. Some acrolein-modified endogenous oligopeptides based on residue–acrolein reactions have been constantly reported. Amongst these reactions, the conjugation of acrolein to GSH is the first step in the acrolein detoxification pathway [43]. Carnosine and homocarnosine are dipeptides that are present at high levels in the brain, and Carini et al. inferred the structures of adducts formed by the reaction cascade of these two dipeptides and acrolein on the basis of MS/MS fragment information [96]. The endogenous tripeptide glycyl–histidyl–lysine (GHK) is a human growth factor, and Giangiacomo et al. characterised its conjugates with acrolein [97]. The structures of macromolecular peptides or proteins obtained through the mechanisms shown in Figure 7A,B can be more complicated due to the multiple modification pathways and modification sites in one molecule. Furthermore, these conjugations can induce protein cross-linking, cyclisation and aggregation which may disrupt the functions of enzymes or functional proteins.

Given that protein adduction is an important mechanism through which acrolein exerts its toxicity, many researchers have paid attention to the modification of specific proteins by acrolein. Some results reported in the last decade (2010–2021) are presented in Table 2 [98,99,100,101,102,103,104,105,106,107,108,109,110,111,112,113,114,115,116,117,118,119]. Acrolein can modify various proteins from all kinds of sources, such as lipoproteins, structural proteins and functional proteins. Acrolein preferentially conjugates to the cysteine, lysine and histidine residues of proteins due to its electrophilic characteristics. These specific adductions can affect the original functions of the target proteins, including protein stability, cytoskeleton maintenance, oxidative stress and immunity. Considering that the conjugated cysteine residue may be the active site, the binding affinity of proteins to their substrates may be impeded, thereby causing functional activation or inactivation. Acrolein modification can be found in many disease models. Although the functional alterations of these target proteins may correlate acrolein with these diseases, additional detailed investigations should be performed to obtain the clear mechanisms through which acrolein-conjugated proteins trigger related diseases. Research methods for identifying the modified sites of proteins still rely on traditional mass spectrometry analysis, which provides limited information. Numerous new technologies, such as genetically encoded sensors [120] and chemoproteomic technology, are being developed and can be applied in the study of protein post-translational modification. Chen et al. established a novel quantitative chemoproteomic method to profile acrolein modification in H1299 cells and identified more than 2300 and 500 acrolein-targeted proteins and cysteine residues, respectively [121]. In conclusion, additional evidence should be obtained to determine the pathological role of acrolein-modified proteins in related diseases, and the combined application of artificial intelligence and biological technology may help make reliable predictions and accelerate the discovery of target proteins with real pathologic importance.

#### 3.2.2. Conjugations of Acrolein to Nucleic Acids

As another intracellular nucleophile, nucleic acids can react with acrolein at the sites of their nitrogenous bases and form many kinds of adducts. In the last few decades, nucleoside–acrolein adducts with different structures were identified consecutively and included all of the bases that constitute DNA or RNA. Commonly, acrolein reacts with either pyrimidines or purines to form adducts with six-membered nitrogen-containing heterocyclic structures, such as the adducts of deoxyguanosine (dG) [122], deoxyadenosine (dA) [123] and deoxycytidine (dC) [124]. The adducts of uridine or thymidine with acrolein exhibit an exocyclic propionaldehyde in their structures [125,126]. Agnieszka et al. suggested that in the case of dA, the addition reaction can be conducted by two molecules of acrolein and results in adducts with bicyclic or FDP structures [127]. In particular, Ivan et al. provided the mode of DNA interchain cross-link formation via the reaction between acrolein and two dG units [128]. The formation mechanism of the adducts of nitrogenous bases and acrolein is shown in Figure 7C.

Amongst these adducts, the adducts of dG (Acr–dG) have been used as biomarkers for the detection of acrolein-induced DNA adducts in cells or tissues. Acr–PdG has two isomers, namely α-OH–Acr-dG and γ-OH–Acr-dG, and a recent study reported that the regioisomer ratio of these adducts is affected by in vivo AAs, proteins and lysates [129]. Not only will the formation of Acr–dG cause DNA damage, but it can also lead to mutations and impair DNA repair capacity [130]. Considerably elevated levels of Acr–dG can be detected in patients with some chronic diseases, including Alzheimer’s disease [131], type 2 diabetes [132] and chronic kidney disease [133]. Given that oxidative stress and lipid peroxidation are more common in tumours than in normal tissues, Acr–dG levels are significantly elevated in samples from patients with cancer. In 2006, Feng et al. found that acrolein preferentially binds to the CpG sites of p53, which is known as a tumour suppressor gene, and contributes to lung carcinogenesis via DNA damage and DNA repair inhibition [134]. Lee et al. proposed that acrolein is a major bladder carcinogen, and Acr–dG might cause further mutations [135]. Similarly, Tsou et al. showed that the levels of Acr–dG in the buccal cells of patients with oral squamous cell carcinoma were 1.4-fold higher than those in the buccal cells of healthy subjects [136]. However, the levels of Acr–dG detected in lung tissues [137] or leucocytes [138] from smokers were not significantly different from those of non-smokers even if acrolein is abundant in cigarette smoke.

The mutagenicity of acrolein–DNA adducts has been confirmed by using a series of shuttle vector-based systems and was discussed by Liu et al. in their previous review [139]. G → T transversions occur most frequently in the mutagenic spectrum [140], and the formation of α-OH–Acr-dG was reported to trigger mispairing with dA and lead to the blockage of polymerase-mediated synthesis or mutagenic effects [141]. Pan et al. suggested that the accumulation of Acr–dG can cause cell apoptosis [142], and Tsai intriguingly found that patients with colorectal cancer and high Acr–dG levels have good prognoses [15]. As inferred from the results of the above studies, even if the genotoxic effects of acrolein–DNA adducts have been confirmed in many models, the real effect of the formation of these adducts in a specific disease remains ambiguous. However, from another perspective, these acrolein–DNA adducts may be applied as novel biomarkers for the early diagnosis, progression assessment and prognosis evaluation of related diseases.

### 3.3. Metabolism and Biotransformation of Acrolein

Acrolein can be metabolised by enzymes and excreted to avoid its long-term retention in the body. Moreover, it can act as an active agent to participate in other metabolic processes and realise its biotransformation. Hence, the metabolism and biotransformation pathways of acrolein are introduced in this section.

Although acrolein has many entry points into the body and harms tissues and organs, it also has numerous metabolic pathways for its clearance and excretion. Richard et al. observed that radioactivity from 2,3-^14^C-labelled acrolein can be found in the expired air (CO_2_), urine, faeces and different tissues of rats after oral or intravenous dosing [143]. Corradi et al. detected strikingly high acrolein content in exhaled breath condensate and sputum from patients with asthma or COPD [144]. Metabolism plays a crucial role in the detoxification of endogenous acrolein, and the major pathways in this process have been described in Steven’s previous review [43]. In brief, acrolein firstly conjugates to GSH in the liver. Secondly, its glutamic acid and glycine residues are cleaved by a specific peptidase. Subsequently, the intermediate is acetylised to form *S*-(3-oxpropyl)-N-acetylcysteine in the kidney. Further oxidation or reduction yields CEMA or HPMA as the major metabolite of acrolein in urine. Richard et al. also proposed the minor metabolic pathways of acrolein in rats [145]. Acrolein can form 3-HPA in the presence of water and then be oxidised stepwise into 3-hydroxypropionic acid, malonic acid and oxalic acid with the simultaneous release of CO_2_. In addition, acrolein may be polymerised to form homopolymers with molecular weights of 2000–20,000 Da in the gastric intestinal tract. These polymers will be excreted with faeces because they are indigestible. The metabolic pathways of endogenous acrolein are illustrated in Figure 8A.

Acrolein can undergo endogenous biotransformation during its participation in the metabolism of other substances (Figure 8B). Spermidine not only generates acrolein during metabolism but also traps acrolein and forms a major adduct with the FDP structure [146]. Ayumi et al. also found that spermine or spermidine can rapidly react with acrolein to form 1,5-diazacyclooctane with an eight-membered ring structure via a [4 + 4] cycloaddition reaction [147]. Although the formation of diazacyclooctane can neutralise the toxicity of acrolein, once acrolein is present in excess, this cycloproduct can form additional toxic gel-like polymers that accelerate the oxidative stress process [148]. Nevertheless, another study found that the formation of 1,5-diazacyclooctane can inhibit Aβ40 peptide fibrillation and significantly reduce cytotoxicity; these results indicated that acrolein and polyamines have potential modulating effects in neural diseases [147].

Zhang et al. elucidated the gut microbial transformation of heterocyclic amines by acrolein (Figure 8C). They suggested that acrolein from the microbial metabolism of glycerol transforms heterocyclic amines into cyclic adducts. For example, they detected the presence of IQ, MeIQx and their acrolein-induced adducts IQ-M1 and MeIQx-M1 during the growth of *Eubacterium hallii* DSM 3353. They also confirmed that acrolein-induced transformation can decrease the cytotoxicity and mutagenicity of MeIQx [149]. In a further study, they observed the similar biotransformation of PhIP into PhIP-M1 by acrolein [150]. Similar to fighting poison with poison, the toxic acrolein mediates the detoxification transformation of those mutagenic heterocyclic amines with the help of gut microbiota.

In summary, acrolein metabolism is an essential approach for decreasing the toxic effects of acrolein. The interference of acrolein during the metabolism of other substances should be noted in addition to the metabolic pathways of acrolein. Acrolein can indirectly modulate cell states via biotransformation with risks and benefits. Dissecting the mechanisms and conditions behind these transformations may establish the relationship between acrolein and related diseases and eventually tip the balance from risks to benefits through some metabolic interventions.

## 4. Concluding Remarks and Future Directions

Acrolein has detrimental effects on health due to its comprehensive sources and high toxicity. Dietary intake is the main exogenous source of acrolein. Given its simple but highly reactive structure, acrolein can rapidly interact with small molecules and macromolecules in foods to form different products. This may re-expose acrolein after food intake and lead to underestimation of its presence based on the level of free acrolein in foods. Even if acrolein scavenging agents can considerably decrease acrolein contents, the adducts formed during these elimination reactions may introduce new risks. In the human body, the conjugations of acrolein to biomacromolecules provide additional detailed pathways through which acrolein triggers diseases. Furthermore, the interference of acrolein in other metabolic processes may indirectly modulate cell states, which will suspend or promote the development of diseases. The mechanisms that drive the origins and fates of acrolein are elaborated in this paper, offering ideas for the effective and safe reduction of acrolein-induced risks and for new treatments targeting acrolein-related diseases.

However, many questions and challenges, especially from the perspective of mechanisms, remain unresolved. Future studies can focus on the following points:

(1) The simple interaction mechanisms found so far between acrolein and single molecules in foods can only explain the disappearance of a small portion of acrolein in foods. Considering that nucleophilic agents, acrolein and other toxic aldehydes co-exist in a real food system, more effort should be invested to discover the simultaneous interaction between acrolein and multiple molecules in foods. 

(2) The risks of product formation from acrolein during elimination reactions should be evaluated to assess their safety, including application of some acrolein-injured models in toxicological and pharmacokinetic studies. 

(3) Given their complicated conversion pathways, the interaction products, especially those with high toxicity, derived from acrolein should be detected to thoroughly estimate real exposure risk. However, acrolein detection is currently limited to the common chemical derivatisation method and ELISA. Therefore, new approaches that determine the levels of acrolein from different sources are urgently needed.

## Figures and Tables

**Figure 1 foods-11-01976-f001:**
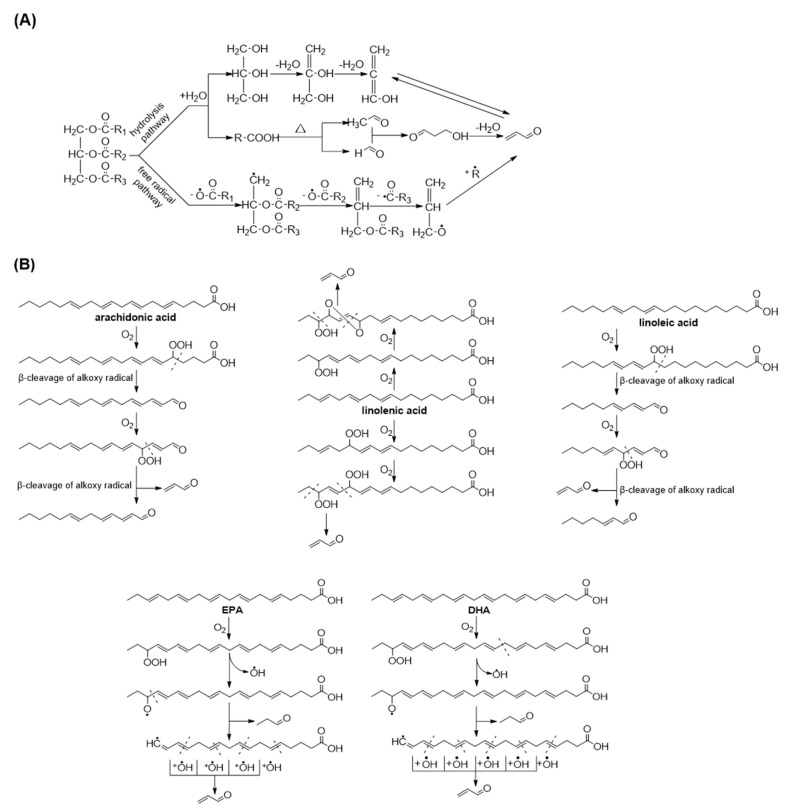
Acrolein generation from lipids. (**A**) Acrolein form triglycerides via hydrolysis and free radical pathways; (**B**) Acrolein from the peroxidation of polyunsaturated fatty acids.

**Figure 2 foods-11-01976-f002:**
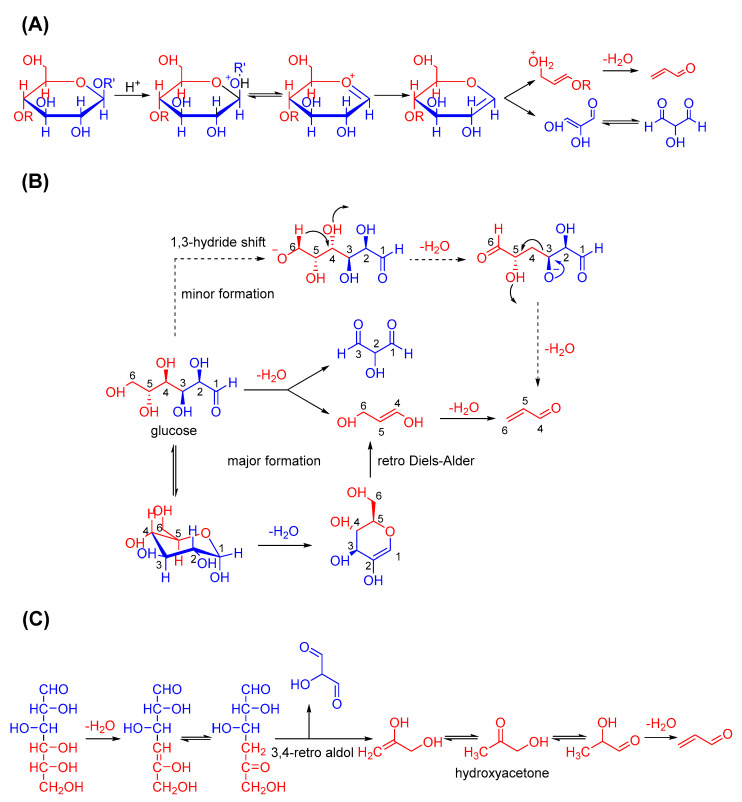
Acrolein formation from carbohydrate pyrolysis. (**A**) Acrolein formation from carbohydrate pyrolysis; (**B**) Acrolein formation from glucose pyrolysis; (**C**) Acrolein formation from hydroxyacetone pathway.

**Figure 3 foods-11-01976-f003:**
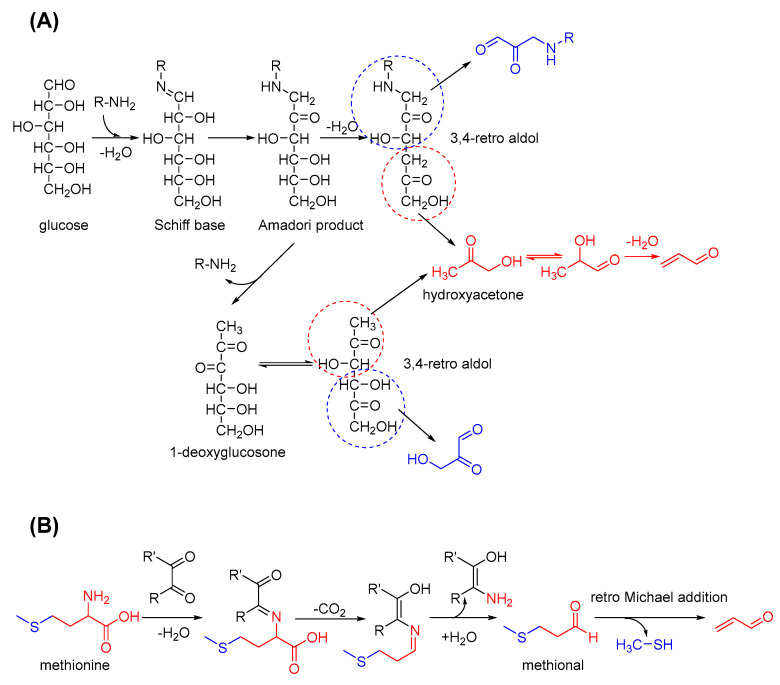
Acrolein formation via the Maillard reaction and Strecker degradation. (**A**) Acrolein formed in the early stage of Maillard reaction: via 3,4-retro aldol reaction; (**B**) Acrolein formed in the intermediate stage of Maillard reaction: via Strecker degradation.

**Figure 4 foods-11-01976-f004:**
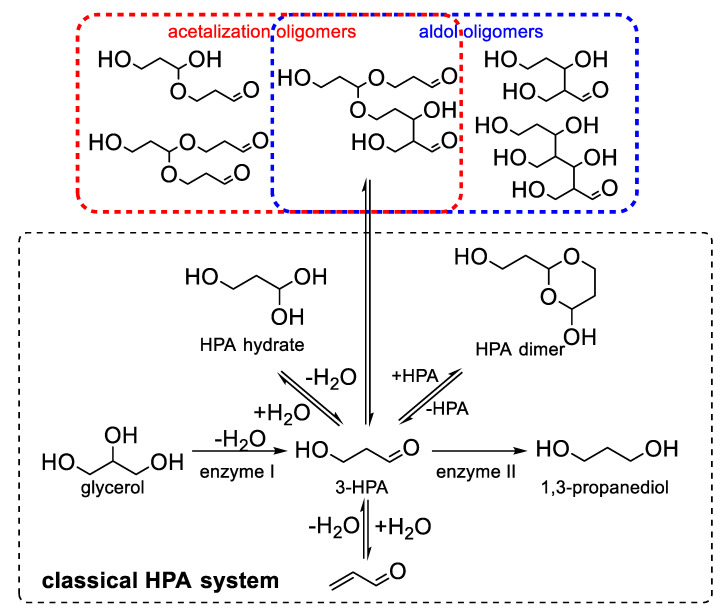
The proposed pathways of acrolein formation in alcoholic beverages by the HPA system.

**Figure 5 foods-11-01976-f005:**
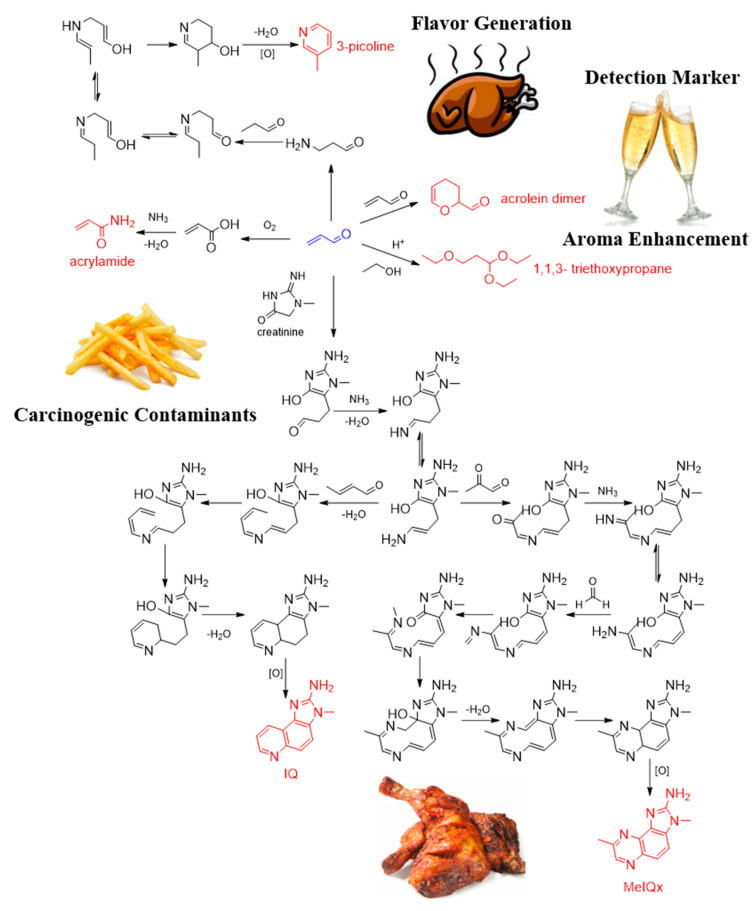
The conversion pathways of acrolein during food processing and storage.

**Figure 6 foods-11-01976-f006:**
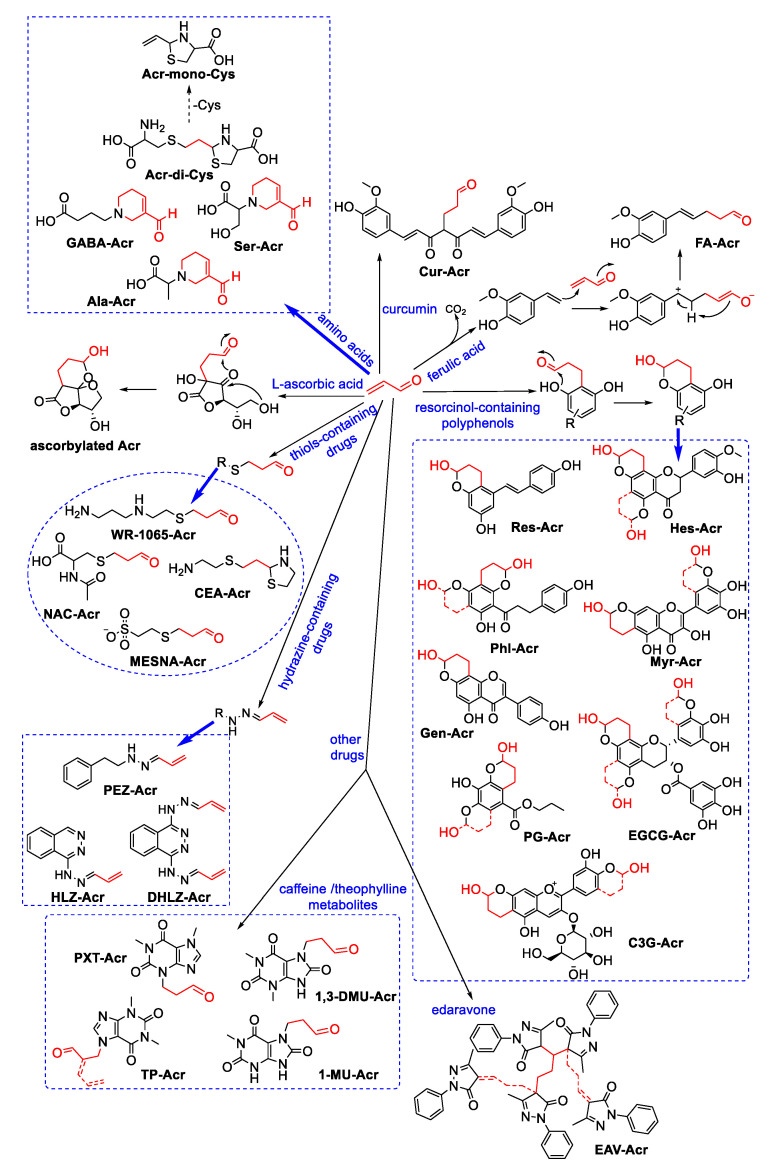
The scavenging mechanism of food components, natural products and clinical drugs on acrolein.

**Figure 7 foods-11-01976-f007:**
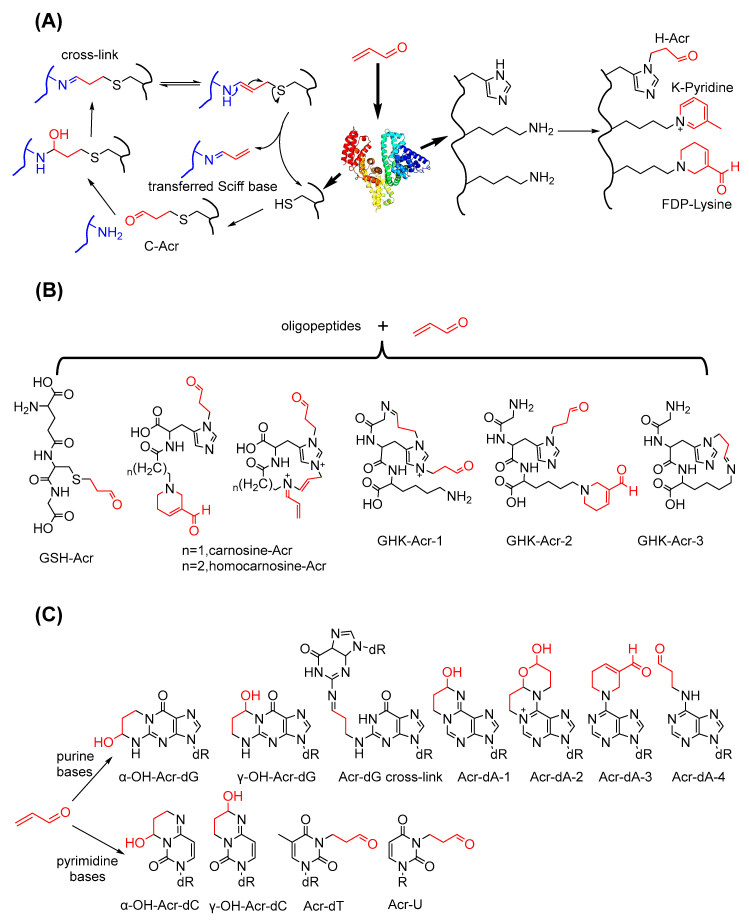
The conjugation mechanisms of acrolein to biomacromolecules. (**A**) Conjugation modes of amino acid residues with acrolein; (**B**) Acrolein-conjugated products of endogenous oligopeptides; (**C**) Acrolein-induced adduction products of nitrogenous bases.

**Figure 8 foods-11-01976-f008:**
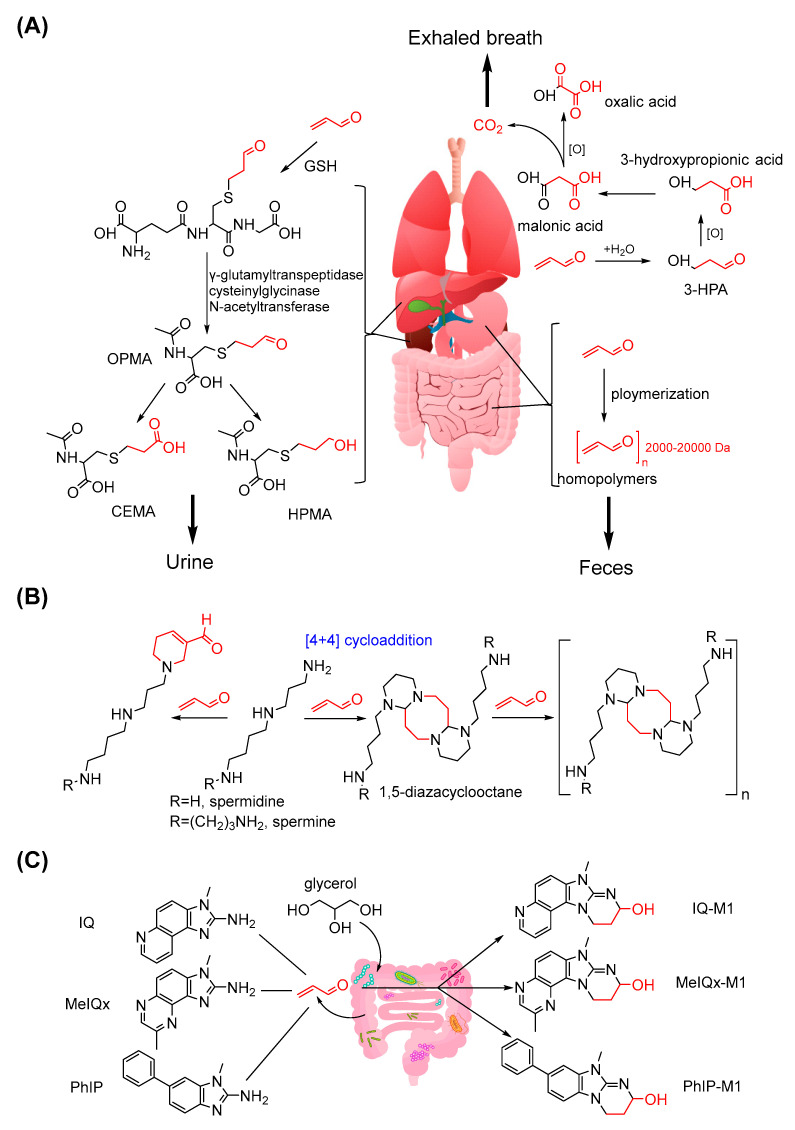
The metabolism pathways and biotransformation of endogenous acrolein. (**A**) Metabolism pathways and metabolites of acrolein; (**B**) Biotransformation of acrolein by reacting with polymaines; (**C**) Biotransformation of acrolein by reacting with heterocyclic amines.

**Table 1 foods-11-01976-t001:** Acrolein content in foods and beverages.

Food	Content (μg/kg or L)	Food	Content (μg/kg or L)
Fruits [21]	10–50	Roasted Cocoa Beans [24]	0.25–0.45
Vegetables [21]	590	Fish Oil [25]	200–1600
Cheese [10]	1000	Frying Oils [26]	7400–198,100
Doughnuts [18]	14.1–16.9	Frying Fats [26]	56,500
Codfish Fillet [10]	100	Cognacs [27]	1420–1500
Sour Dough [22]	14.72	Scotch Whiskey [28]	670–11,100
Bread [22]	161	Sparkling Wine [29]	20.3–33.4
French Fries [18]	14.8–19.9	Red Wine [30]	1.0–1.5
Potato Chips [18]	16.3–23.3	Cider [31]	2600–31,800
Frying Cassava [23]	1.7–10.2	Beer [32]	<2.5–5.4
Frying Pork Sausage [23]	≈2–6		

**Table 2 foods-11-01976-t002:** Protein modifications induced by acrolein.

Sample	Target Protein	Modified Site	Interfered Function	Related Disease	Ref.
Plasma, patients	albumin	K557, K560	-	Silent brain infarction	[98]
In vitro reaction	recombinant JNK2α2	C41, C177	Weaken interations with MKK4/7	-	[99]
In vitro reaction	rat apolipoprotein E	M60, K64, K68, K135, K138, K149, K155, K254	Impair plasma cholesterol homeostasis	Dysregulation in lipid metabolism	[100]
Neuro2a cells	GAPDH	C150, C282	Inactivation of GAPDH	-	[101]
Plasma, patients	albumin	C34	Oxidative stress	Ischemia-reperfusion injury	[102]
HBE cells, human lung explants	cystic fibrosis transmembrane conductance regulator	C524, C647, C1395, K464, K1334, K1177, K532	Instability at the cell suface	Chronic obstructive pulmonary disease	[103]
U-937 monocytes	hinder histone deacetylases 2	C274	-	Chronic obstructive pulmonary disease	[104]
BEAS-2B cells	histone H4	K5, K8, K12, K16	Chromatin assembly	-	[105]
Lung tissues, C57BL/6 mice	surfactant protein A	H39, H116, C155, K180, K221, C224	Immunity dysfunction	Chronic obstructive pulmonary disease	[106]
Saliva, patients	MMP-9	C99, C230, C244, C302, C314, C329, C347, C361, C388, K384, H405, H411, C516, K535	Activation of MMP-9	Primary Sjögren’s syndrome	[107]
In vitro reaction	lysozyme	C6, C30, C64, C76, C80, K96, C155, K116	-	-	[108]
In vitro reaction	human serum albumin	C34, H67, K137, H146, K262, K276, H288, H338, K414, K525, K574	-	-	[108]
In vitro reaction	short palate lung and nasal epithelial clone 1	C180, C224	Disruption of the disulfide bond	Chronic obstructive pulmonary disease	[109]
Brain tissue, SD rat	α-synuclein	-	Oligomerisation and aggregation	Parkinson’s disease	[110]
In vitro reaction	α- and β-tubulins	C25, C295, C347, C376 in α-tubulin; C12, C129, C211, C239, C303, C354 in β-tubulin	Microtubule formation inhibition	Brain infarction	[111]
bEnd.3 cells	apolipoprotein E3	K1, K69, K72, K75, K95, K157, K233, K242, K282	Impair plasma cholesterol homeostasis	Lipid disorders	[112]
MCF-7 cells	pyruvate kinase	C49, C152, K166, K207, C358, H391, K393, C423, C474, K475	Activity inhibition	Cancer	[113]
Lung tissues, C57BL/6 mice	surfactant protein D	K243, K246, K287, K299, K303	Immunity dysfunction	Smoking-associated respiratory diseases	[114]
Saliva, patients	immunoglobulins	K43(λ); K75, K80, H81, K82, C86(κ); C300(α-2); C27, K30(γ-1); C297, K300 (γ-3)	Autoantibody functionalisation	Primary Sjögren’s syndrome	[115]
Epithelial lining fluid, C57BL/6J mice	albumin	C34	HMOX1 transcript increase	-	[116]
FM3A cells	vimentin	C328	Dysfunction of the cytoskeleton	Brain infarction	[117]
FM3A cells	actin	C217, C257, C285, K118	Dysfunction of the cytoskeleton	Brain infarction	[117]
Low density lipoprotein	apolipoprotein B-100	C212, K327, K742, K949, K1087, H1923, K2634, K3237, K3846	Uptake of LDL	-	[118]
Hippocampus tissues, C57BL/6 mice	14-3-3 protein	-	Aggregation of tau	Alzheimer’s disease	[119]

## Data Availability

Not applicable.

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
