# Peer review of "Origin and Fate of Acrolein in Foods"

_foods, 2022, doi:10.3390/foods11131976_

Round 1

Reviewer 1 Report

The manuscript entitled "Origin and fate of acrolein: a review with a focus on molecular mechanisms" introduces and describes in a discursive and, to some extent, almost elementary way. The review is generally lacking in well-organised and described structure. As well as, there does not seem to be any connection between the various topics covered, but a simple list of the various focuses. Furthermore, the file uploaded for the draft version does not include the figures and tables mentioned in the paper.

The paper need to be intensively revised, and also a major editing of english is required, paying attention to the use of some strong terms (es. lucubrated, "the last but not the least", erratically, victim, routes, sacrificially, exacerbated, etc.)

Lines 13-19: The abstract is very simple and is nothing more than a list of the topics that will be covered. It should be reworded in a way that is more interconnected to the focus and also briefly describe the results of the study.

Line 24: Is acrolein only derived from soap manufacture?

Line 27: "acrolein is the simplest..." you should not use such categorical or absolute terms. You should also added references.

Lines 27-48: The information described in this part does not follow a logical thread. They result as a series of cascade information. It would be good to organise it in a well-organised summary figure, presenting in an introductive way, the main causes, the health effects, and references.

Line 44: "lucubrated"? What does it stand for? It is a little bit strong.

The paper lacks a section on materials and methods, used in the writing of the review. In the absence of this, the paper is nothing more than a list of topics already found in the literature. 

Line 50-328: This whole part about the origin of the molecule and its different sources is really too long-winded. Furthemore, data are presented randomly. All this part related to the origin and sources should be elaborated. It should be better organised in a summary table (in which origine, different sources, concentration are showed).  Then, all the data showed in the table should be discussed, elaborated and argued with a logical interconnection.

Lines 297-299: This sentence is so obvious that there is no point in writing it.

Also the paragraph 2 about "Fate of acrolein" is still prolix and seems to be an in-depth summary of the literature, instead of being a revision. Many concepts are redundant in the text, with no linkage within them. It would be good to re-organize it.

Author Response

The manuscript entitled "Origin and fate of acrolein: a review with a focus on molecular mechanisms" introduces and describes in a discursive and, to some extent, almost elementary way. The review is generally lacking in well-organised and described structure. As well as, there does not seem to be any connection between the various topics covered, but a simple list of the various focuses. Furthermore, the file uploaded for the draft version does not include the figures and tables mentioned in the paper.

Answer: The figures and tables mentioned in the paper have been attached at the end of the manuscript.

The paper need to be intensively revised, and also a major editing of english is required, paying attention to the use of some strong terms (es. lucubrated, "the last but not the least", erratically, victim, routes, sacrificially, exacerbated, etc.)

Answer: Thank you for your kind suggestion. The revised manuscript has been adjusted to focus on acrolein in foods, and the language has been edited by a native speaker (Essaystar.com)

Lines 13-19: The abstract is very simple and is nothing more than a list of the topics that will be covered. It should be reworded in a way that is more interconnected to the focus and also briefly describe the results of the study.

Answer: Thank you for your kind suggestion. The abstract has been reised to be more interconnected to the focus of acrolein in foods.

Line 24: Is acrolein only derived from soap manufacture?

Answer: Acrolein can be also derived from fuel combustion, food processing, drug metabolism in human body and many other ways. We mentioned the soap manufacturing industry to introduce the discovery process of acrolein from glycerol in the history. Thank you.

Line 27: "acrolein is the simplest..." you should not use such categorical or absolute terms. You should also added references.

Answer: Thank you for your kind suggestion. The sentence has been rewritten to be more precise as “Acrolein is the structurally simplest α, β-unsaturated aldehyde….”, and the reference has been added.

Lines 27-48: The information described in this part does not follow a logical thread. They result as a series of cascade information. It would be good to organise it in a well-organised summary figure, presenting in an introductive way, the main causes, the health effects, and references.

Answer: Thank you for your kind suggestion. Some new contents about acrolein in foods has been supplemented in the introduction section to make it more concentrated on the topic and emphasize the necessity to investigate sharp elimination of acrolein after its formation in foods.

Line 44: "lucubrated"? What does it stand for? It is a little bit strong.

Answer: We rewrote the introduction section and this word has been deleted in the revised version. Thank you.

The paper lacks a section on materials and methods, used in the writing of the review. In the absence of this, the paper is nothing more than a list of topics already found in the literature.

Answer: Thank you for your suggestion. We deleted some contents and only retained the part of acrolein in foods to emphasize this topic in the revised version.

Line 50-328: This whole part about the origin of the molecule and its different sources is really too long-winded. Furthemore, data are presented randomly. All this part related to the origin and sources should be elaborated. It should be better organised in a summary table (in which origine, different sources, concentration are showed).  Then, all the data showed in the table should be discussed, elaborated and argued with a logical interconnection.

Answer: Thank you for your suggestion. We focused on acrolein in foods added subtitles to give the topic discussed in each subsection.

Lines 297-299: This sentence is so obvious that there is no point in writing it.

Answer: The sentence has been deleted in the revised version. Thank you.

Also the paragraph 2 about "Fate of acrolein" is still prolix and seems to be an in-depth summary of the literature, instead of being a revision. Many concepts are redundant in the text, with no linkage within them. It would be good to re-organize it.

Answer: Thank you for your suggestion. We have re-organized this section and retained the content that focus on acrolein in foods and human body. New subtitles were also added to clear the topic discussed in each subsection.

Reviewer 2 Report

The paper provides a substantial contribution to the sources/exposure routes of aclolein and the molecular explanations regarding the fate and effects of acrolein in living systems.

However, I have raised some issues and suggested some changes that I think are necessary. Unconventional acronyms were used without first defining them. Sectional and sub-sectional numbering systems are incoherent. I couldn't find any of the Figures and Tables mentioned in this paper in the version of the manuscript I down from the link sent to me. I feel the authors can do more to emphasize the toxicological effects of aclolein in foods and suggest ways of ameliorating these effects in human foods and in the body; considering that the paper is to be published in foods (MDPI) if accepted. Otherwise, they could consider transferring the manuscript to a sister journal, molecules (MDPI), in its current form; after effecting the required changes.

Comments for authors - foods-1755139

1.      Title: It will be good if the authors can indicate the type of review being presented to the readership. Are you doing a scoping, systematic or meta-analytical review? For instance, you could edit the title to read “Origin and fate of acrolein: a scoping review ...”

2.      Abstract: Line 13: What is “ambient atmosphere? The word “ambient” makes it difficult to understand the sources of acrolein being listed. I suggest that the authors delete the word “ambient” so that the sentence may read “… industrial production, the atmosphere, …”

3.      Keywords:  Although not compulsory, it will be nice to have the keywords listed in alphabetical order.

4.      Introduction: Line 24: …. a by-product of soap manufacturing industry,...?

5.      Line 42:  What is IPCS? I guess the authors are referring to International Programme on Chemical Safety? If so, they should statutorily state the full meaning and then use the acronym subsequently all through the manuscript.

6.      Lines 49-57: Please look at your numbering pattern in the sections and subsections. There should be no 2.1 (Industrial sources of acrolein) when “the origins of acrolein” that was supposed to be numbered 2 wasn’t numbered. So, assign number 2 to the main section, Origins of acrolein in line 49.

7.      Line 99: EPA???

8.      Line 100: What exactly do "Ambient" mean here? Ambient air (atmosphere) or ambient temperature? This will guide the authors to recast the sentence accordingly for a better intellectual meaning. I think the sentence should read “ The atmospheric concentrations of acrolein differ across diverse regions…”

9.      Line 112: What isyoung” fire smoke? Do you mean recent fire smoke?

10.  Line 127: Qian et al. Authors should provide a customary citation number in square bracket and reference the citation accordingly at the reference list.

11.  Line 182 and elsewhere in the manuscript: References were made to tables and figures but I couldn’t see these tables and figures in the version of the manuscript sent to me. Ideally, tables and figures should be placed as close as possible to the places where they were mentioned in the manuscript. This will help the readership to understand or confirm your claims.

12.  Line 195: I suggest that the authors insert a sub-section/heading “mechanisms of acrolein generation in food ” in line 195.

13.  Lines 215-116: Please define the acronym DHA first before using it. You can not take for granted that everybody know the full meaning of the acronym as relates to foods.

14.  Line 330: “2. Fate of acrolein” Another wrong numbering system

15.  Line 372: IARC???

16.  Line 600: “3. Concluding remarks and future directions” – Numbering issue

Author Response

The paper provides a substantial contribution to the sources/exposure routes of aclolein and the molecular explanations regarding the fate and effects of acrolein in living systems.

However, I have raised some issues and suggested some changes that I think are necessary. Unconventional acronyms were used without first defining them. Sectional and sub-sectional numbering systems are incoherent. I couldn't find any of the Figures and Tables mentioned in this paper in the version of the manuscript I down from the link sent to me. I feel the authors can do more to emphasize the toxicological effects of aclolein in foods and suggest ways of ameliorating these effects in human foods and in the body; considering that the paper is to be published in foods (MDPI) if accepted. Otherwise, they could consider transferring the manuscript to a sister journal, molecules (MDPI), in its current form; after effecting the required changes.

Comments for authors - foods-1755139

  1. Title: It will be good if the authors can indicate the type of review being presented to the readership. Are you doing a scoping, systematic or meta-analytical review? For instance, you could edit the title to read “Origin and fate of acrolein: a scoping review ...”

Answer: Thank you for your kind suggestions. We have adjusted the title as “Origin and fate of acrolein in foods” to emphasized our focus.

  1. Abstract: Line 13: What is “ambient atmosphere? The word “ambient” makes it difficult to understand the sources of acrolein being listed. I suggest that the authors delete the word “ambient” so that the sentence may read “… industrial production, the atmosphere, …”

Answer: Thank you for your suggestion. We deleted the word “ambient” and reworded the abstract to be more interconnected to the focus of acrolein in foods.

  1. Keywords: Although not compulsory, it will be nice to have the keywords listed in alphabetical order.

Answer: Thank you for your suggestion. The keywords have been listed in alphabetical order.

  1. Introduction: Line 24: …. a by-product of soap manufacturing industry,...?

Answer: Thank you for your suggestion. The sentence has been corrected.

  1. Line 42: What is IPCS? I guess the authors are referring to International Programme on Chemical Safety? If so, they should statutorily state the full meaning and then use the acronym subsequently all through the manuscript.

Answer: Thank you for your suggestion. The full meanings of these acronyms have been stated in the revised version.

  1. Lines 49-57: Please look at your numbering pattern in the sections and subsections. There should be no 2.1 (Industrial sources of acrolein) when “the origins of acrolein” that was supposed to be numbered 2 wasn’t numbered. So, assign number 2 to the main section, Origins of acrolein in line 49.

Answer: Thank you for your suggestion. The numbering pattern has been assigned in the revised version.

  1. Line 99: EPA???

Answer: Thank you for your suggestion. The full meaning of EPA here is Environmental Protection Agency and has been placed instead of its acronym in the revised version.

  1. Line 100: What exactly do "Ambient" mean here? Ambient air (atmosphere) or ambient temperature? This will guide the authors to recast the sentence accordingly for a better intellectual meaning. I think the sentence should read “ The atmospheric concentrations of acrolein differ across diverse regions…”

Answer: Thank you for your suggestion. This part has been deleted in the revised version, and we retained the content about acrolein in foods to highlight the topic.

  1. Line 112: What is “young” fire smoke? Do you mean recent fire smoke?

Answer: Thank you for your suggestion. This part has been deleted in the revised version, and we retained the content about acrolein in foods to highlight the topic.

  1. Line 127: Qian et al. Authors should provide a customary citation number in square bracket and reference the citation accordingly at the reference list.

Answer: Thank you for your suggestion. This part has been deleted in the revised version, and we retained the content about acrolein in foods to highlight the topic.

  1. Line 182 and elsewhere in the manuscript: References were made to tables and figures but I couldn’t see these tables and figures in the version of the manuscript sent to me. Ideally, tables and figures should be placed as close as possible to the places where they were mentioned in the manuscript. This will help the readership to understand or confirm your claims.

Answer: We are sorry that we didn’t successfully upload the tables and figures in the initial version. The tables and figures have been uploaded and placed in the revised manuscript. Thank you for your suggestion.

  1. Line 195: I suggest that the authors insert a sub-section/heading “mechanisms of acrolein generation in food ” in line 195.

Answer: Thank you for your kind suggestion. We added new subtitles in the revised version as you suggested to clear the topic discussed in each subsection.

  1. Lines 215-116: Please define the acronym DHA first before using it. You can not take for granted that everybody know the full meaning of the acronym as relates to foods.

Answer: Thank you for your suggestion. The full meanings of EPA and DHA here have been stated before using their acronyms in the revised version.

  1. Line 330: “2. Fate of acrolein” Another wrong numbering system

Answer: Thank you for your suggestion. The numbering pattern has been corrected in the revised version.

  1. Line 372: IARC???

Answer: IRAC means International Agency for Research on Cancer and the full meaning has been stated in the revised version. Thank you.

  1. Line 600: “3. Concluding remarks and future directions” – Numbering issue

Answer: Thank you for your suggestion. The numbering pattern has been corrected in the revised version.

Reviewer 3 Report

The manuscript submitted for review deals with acrolein, which is a highly reactive compound and highly toxic to aquatic organisms. There are studies on the toxic effects of acrolein on the aquatic environment due to its use as a herbicide in irrigation canals. The article is interesting, with a few suggestions/feedbacks to the prepared content that should have been taken into consideration, which are presented below:

Line 25, 26, 151, 153, 251, 322: spaces should be filled in, in addition, I ask the Authors of the manuscript to carefully check the journal's requirements for word processing, which the available template helps with.

Line 46: it would be worth noting on which journal databases the review article was based, especially as it is very large (43 pages), which should also be analyzed and considered. In addition, the tables and figures that the authors refer to in the text of the manuscript are not included, I did not find them.

Line 49: Chapter number is missing, it should be 2. Origin of acrolein

Line 62: I propose 300-320 °C

Line 99, 310: please explain the abbreviation when first used in the text: EPA, AAs

Line: 330: should be 3. Fate of acrolein

Line 600: Should be chapter 4, not 3

The article needs corrections and additions to match the journal's requirements.

Author Response

Reviewer 3

The manuscript submitted for review deals with acrolein, which is a highly reactive compound and highly toxic to aquatic organisms. There are studies on the toxic effects of acrolein on the aquatic environment due to its use as a herbicide in irrigation canals. The article is interesting, with a few suggestions/feedbacks to the prepared content that should have been taken into consideration, which are presented below:

Line 25, 26, 151, 153, 251, 322: spaces should be filled in, in addition, I ask the Authors of the manuscript to carefully check the journal's requirements for word processing, which the available template helps with.

Answer: Revised as you suggested. Thank you.

Line 46: it would be worth noting on which journal databases the review article was based, especially as it is very large (43 pages), which should also be analyzed and considered. In addition, the tables and figures that the authors refer to in the text of the manuscript are not included, I did not find them.

Answer: Thank you for your suggestion. We deleted some contents and only retained the part of acrolein in foods to emphasize this topic in the revised version. We are sorry that we didn’t successfully upload the tables and figures in the initial version. The tables and figures have been uploaded and placed in the revised manuscript as you suggested.

Line 49: Chapter number is missing, it should be 2. Origin of acrolein

Answer: Thank you for your suggestion. The numbering pattern has been added in the revised version.

Line 62: I propose 300-320 °C

Answer: Thank you for your suggestion. We deleted this part and retained the content about acrolein in foods to emphasized the topic.

Line 99, 310: please explain the abbreviation when first used in the text: EPA, AAs

Answer: Thank you for your suggestion. The full meanings of these acronyms have been stated in the revised version, and “AAs” in the initial version has been replaced with AA which means amino acids.

Line: 330: should be 3. Fate of acrolein

Answer: Thank you for your suggestion. The numbering pattern has been added in the revised version.

Line 600: Should be chapter 4, not 3

Answer: Thank you for your suggestion. The numbering pattern has been added in the revised version.

Round 2

Reviewer 1 Report

The review has been improved according to the reviewers suggestions. Therefore I lastly suggest to insert the figures in the text and not at the end of the manuscript. Also the Table 2. should be formatted in a horizontal page.

Reviewer 2 Report

The paper has been significantly improved.